physiology, ecology

symbiosis, nutrient limitation, photosynthesis, Symbiodiniaceae, mutualism, NanoSIMS

**Authors for correspondence:**
Thomas Krueger
e-mail: tk556@cam.ac.uk
Anders Meibom
e-mail: anders.meibom@epfl.ch

†Present address: Department of Biochemistry, University of Cambridge, Cambridge CB2 1QW, UK.

# Intracellular competition for nitrogen controls dinoflagellate population density in corals

Thomas Krueger[1,†], Noa Horwitz[2,3], Julia Bodin[1], Maria-Evangelia Giovani[1], Stéphane Escrig[1], Maoz Fine[2,3] and Anders Meibom[1,4]

[1]Laboratory for Biological Geochemistry, School of Architecture, Civil and Environmental Engineering, Ecole Polytechnique Fédérale de Lausanne (EPFL), 1015 Lausanne, Switzerland
[2]The Mina and Everard Goodman Faculty of Life Sciences, Bar-Ilan University, Ramat-Gan 52900, Israel
[3]The Interuniversity Institute for Marine Sciences, Eilat 88103, Israel
[4]Center for Advanced Surface Analysis, Institute of Earth Sciences, University of Lausanne, 1015 Lausanne, Switzerland

TK, 0000-0002-8132-8870

The density of dinoflagellate microalgae in the tissue of symbiotic corals is an important determinant for health and productivity of the coral animal. Yet, the specific mechanism for their regulation and the consequence for coral nutrition are insufficiently understood due to past methodological limitations to resolve the fine-scale metabolic consequences of fluctuating densities. Here, we characterized the physiological and nutritional consequences of symbiont density variations on the colony and tissue level in *Stylophora pistillata* from the Red Sea. Alterations in symbiont photophysiology maintained coral productivity and host nutrition across a broad range of symbiont densities. However, we demonstrate that density-dependent nutrient competition between individual symbiont cells, manifested as reduced nitrogen assimilation and cell biomass, probably creates the negative feedback mechanism for symbiont population growth that ultimately defines the steady-state density. Despite fundamental changes in symbiont nitrogen assimilation, we found no density-related metabolic optimum beyond which host nutrient assimilation or tissue biomass declined, indicating that host nutrient demand is sufficiently met across the typically observed range of symbiont densities under ambient conditions.

## 1. Introduction

Tropical reef-building scleractinian corals are unique symbiotic organisms that involve at least three major groups: the cnidarian host animal, microalgal dinoflagellates of the family Symbiodiniaceae and the microbial community. The photosynthetic algae inside the tissue are the primary driver of coral productivity and a crucial component for the foundation for reef ecosystems. Immobilized inside arrested host phagosomes [1,2], they assimilate nutrients derived from seawater and the coral's metabolism [3,4], and transfer organic photosynthates to the host animal. This creates an efficient mechanism for the recycling and conservation of nutrients, giving symbiotic corals an ecological advantage in often highly oligotrophic tropical waters.

The coral animal relies on phototrophic carbon input (mainly in the form of sugars and lipids) from its symbionts for sustenance. The release of organic carbon from the dinoflagellate symbiont to the host is essentially based on Sprengel/Liebig's law of the minimum (i.e. the principle that it cannot allocate all photosynthetically fixed carbon into building proteins and cellular structure due to the low availability of required nitrogen and phosphorus [5,6]). The lack of a balanced inorganic nutrient supply to meet the Redfield ratio (the average

composition of phytoplankton biomass with regard to C, N and P) [7] essentially disrupts the symbiont's investment of photosynthetic carbon into its own cell and population growth and triggers its release. The underlying nutrient limitation is thus the core of the coral symbiosis, because it allows the coral animal to harvest released 'excess' phototrophic carbon from its endosymbionts [5,6].

A large body of experimental work has supported the initial hypothesis that coral symbionts *in hospite* are nitrogen-limited [8] and that their population size is primarily regulated by nitrogen availability [9,10]. Elevated nutrient conditions (mainly N, P and some trace metals such as Fe) and/or heterotrophic feeding has been shown to stimulate symbiont cell division, the number of symbionts hosted by one animal cell and the overall number of symbionts per unit of coral surface area in many (but not all) studies [11–22]. Whether and to what extent the steady-state symbiont density is regulated by host-induced nutrient limitation [23] or primarily driven by ambient nutrient/prey availability is still debated. A recent meta-analysis furthermore highlights that the specific form and ratio of ambient inorganic nitrogen and phosphorus are also critical factors [24].

Corals naturally display high variability in their symbiont densities between individuals of the same species and across the surface of individual colonies [10,25,26]. Variations in steady-state symbiont density are governed by a combination of parameters such as tissue thickness, space availability within the tissue, nutrient supply and the rates of symbiont cell division, senescence, expulsion and digestion [10]. Disentangling the interactions and feedbacks that exist between symbiont and host physiology as result of local variations in density is technically challenging, because classical methods yield physiological variables as highly averaged values. Usually, reported cell-normalized values are derived from whole fragment measurements that integrate the signal from tens of millions of cells. Such measurements include oxygen metabolism at the level of an entire colony/fragment in respiration chambers [27,28], light utilization based on centimetre-scale spot PAM fluorometry [29], and bulk tissue measurements for coral biometrics and biochemistry after separating both partners through centrifugation [30]. Likewise, quantifying the mutual nutrient exchange within the coral symbiosis using stable and radioactive isotopes of carbon and nitrogen has been traditionally investigated through bulk analysis [31–36].

Due to methodological limitations, the direct feedbacks between tissue symbiont density, nutrient accessibility and assimilation, and transfer of photosynthates to the host are insufficiently resolved at the local tissue level. Recently, correlated electron microscopy and nanoscale secondary ion mass spectrometry (NanoSIMS) has enabled the tracking and quantitative visualization of elemental and stable isotopic distributions in preserved intact tissue samples with subcellular resolution [37]. This technique has subsequently provided detailed information on the fate of assimilated nutrients in the coral symbiosis [37–40] and has begun to shed light on the importance of the local microenvironment for individual symbiont performance [41]. In addition, NanoSIMS analysis permits *in hospite* measurements of nutrient utilization while simultaneously linking single-cell performance to its local density context in the host tissue. Here, we combined bulk physiological measures across colonies with varying symbiont densities with NanoSIMS isotopic imaging to revisit the physiological and nutritional consequences of symbiont density variations for the coral holobiont on the colony- and tissue-scale level. By comparing corals that were either heterotrophically supplemented by regular feeding or remained unfed, we also investigated whether an enlarged symbiont population amplifies potential density-induced limitations for photosynthesis and symbiont nutrition and how such changes relate to host acquisition of photosynthates. We demonstrate a link between nutrient assimilation and local tissue symbiont density that has direct implications for the population control of dinoflagellates in symbiotic corals.

## 2. Methods

The coral samples, the experimental treatments and the physiological analytical protocols for this study are those described in Krueger *et al.* [40,42]. Physiological and NanoSIMS data from the ambient temperature treatment of both publications were reanalysed when we discovered the density effects described here. Note that due to the comparably high heat tolerance of Northern Red Sea *Stylophora pistillata* [42], which is not characteristic for the typical heat response of most other corals, we only considered the samples from the ambient temperature dataset here.

### (a) Experimental set-up

Paired fragments of nine *Stylophora pistillata* colonies were maintained at ambient conditions (22–24°C, PAR: 300–400 µmol m$^{-2}$ s$^{-1}$) in outdoor flow-through seawater aquaria (the 'Red Sea Simulator' at the InterUniversity of Marine Sciences, Eilat, Israel) for 67 days, either unfed or fed twice a week with 2500 *Artemia* nauplii per fragment and three replicate tanks per treatment with separate replicate colonies in each tank. All colonies were subsequently tested for photosynthetic performance (respirometry + chlorophyll PAM fluorometry) and analysed for their biometric variables (symbiont density, chlorophyll, carbohydrate and protein content), as described previously [42]. For NanoSIMS measurements of nutrient assimilation, fragments of three of the nine mother colonies were incubated in natural seawater with added 2 mM NaH$^{13}$CO$_3$ (98 atom %) and 3 µM K$^{15}$NO$_3$ (98 atom %) for 6 h in the light. Samples were then fixed, decalcified, embedded in Spurr resin and microtomed to measure $^{13}$C and $^{15}$N enrichments in different tissue compartments of coral sections, following established procedures [40].

### (b) NanoSIMS image analysis for density effects

Imaging tissue sections with NanoSIMS creates two-dimensional maps of element and stable isotope distribution and can be used to visualize relative isotopic enrichment in biological samples [43], yielding an assessment of the compartment-specific enrichment *in hospite*. Previously, these images have been exclusively used to determine the $^{13}$C and $^{15}$N enrichment of coral symbiont and host tissue after an isotopic pulse and to investigate subcellular features [37–41,44–47]. However, if a sufficiently high number of identical frame-sized images are obtained that contain a random number of symbionts from across the connective coenenchyme tissue between polyps of a coral colony, these cross-sectional images can be used to investigate how variation in the observed local-scale symbiont density is linked to the nutrient assimilation in both partners (electronic supplementary material, figure S1). For this study, we reanalysed our ambient autotrophy and heterotrophy NanoSIMS dataset from three independent colonies of *Stylophora pistillata* [40] with a specific focus on the effect of local symbiont density variations. The dataset consists of a 64 (autotrophy, acclimated to unfed or fed regimes) and 34 (heterotrophy, fed regime) 40 × 40 µm images, containing

a total of 304 and 138 symbiont cells and their immediate surrounding coral gastrodermal tissue, respectively.

Using L'IMAGE software (Dr Larry Nittler, Carnegie Institution of Washington), regions of interest (ROIs) were defined within each image, yielding multiple symbiont ROIs and one host gastrodermis ROI per image. The number of visible symbionts per image ranged from 1 to 12. L'IMAGE image analysis yielded average $^{13}$C- and $^{15}$N-enrichment as well as size data for each ROI. The specific local gastrodermal symbiont density in each picture was quantified as number of symbiont cells per 500 $\mu m^2$ total cross-sectional area according to

$$\frac{\text{number of symbionts}}{\text{gastrodermal area } [\mu m^2]} \times 500.$$

All enrichment data are expressed as atom per cent excess (APE) [48] and corrected for the initial enrichment level in the labelled seawater or the zooplankton to allow meaningful comparison between both feeding modes [40]. Assuming steady-state conditions of the C and N pool (i.e. assuming a negligible change in total biomass in the imaged tissue areas within the pulse period), the reported normalized APE values in per cent are a measure of the relative structural C and N turnover in each ROI over 6 h in the light.

## (c) Statistical analysis
### (i) Coral physiological data
Physiological effects of regular feeding on the nine colonies were tested via a paired analysis of variance (ANOVA) to test for a treatment difference between paired fragments (i.e. pieces of the same colony went into both treatments) due to regular feeding. The ANOVA included the factors 'feeding', 'replicate [nested in replicate tank]', and 'replicate tank' to account for the split design and the treatment replication [42]. In addition to the mean difference between feeding treatments, we tested whether colony density affects the measured physiological variables and whether feeding state of the holobiont alters this density relationship, using an analysis of covariance (ANCOVA). The ANCOVA included symbiont 'density' as predictor, each physiological variable as response, 'feeding' treatment as interactive cofactor, and 'replicate tank' as another cofactor. Note that the ANCOVA was only used to test for the presence or the absence of density effects, not to quantify them. The effect of density was quantified for the relationship between symbiont density and average soluble symbiont protein and carbohydrate content on the colony level, using orthogonal regression analysis. In contrast to traditional ordinary least-squares (OLS) regression, orthogonal regressions aim to minimize deviations with respect to both x and y in cases where no clear causation can be attributed and both variables have an inherent measurement error.

### (ii) NanoSIMS data
The effect of symbiont density on individual symbiont C and N assimilation on the local tissue level was tested with a factorial linear regression model containing the fixed factors 'feeding' (nominal) and 'symbiont density' (continuous) plus their interaction. Due to the paired (all coral colonies experienced all treatments) and nested (multiple NanoSIMS ROI data points per replicate) nature of the data, the random factors 'replicate' and 'replicate × feeding' were included. Assumptions of multivariate normality (by Shapiro–Wilk test on variable), the absence of multicollinearity (variance inflation factor values), lack of autocorrelation (residual by row plot) and homoscedasticity (residuals versus predicted values plot) were checked. Due to the isotopic hotspot effect on mean cell enrichment (see electronic supplementary material, methods), the NanoSIMS dataset for symbiont C and N assimilation was reduced to only include assimilation data from cells

larger than 5 μm for nitrogen and larger than 6 μm for carbon, while maintaining the local symbiont density seen on each image. This reduced dataset showed no significant effect of symbiont diameter on isotopic enrichment value (electronic supplementary material, figure S2). All statistical analyses were conducted in JMP v. 11.2.1 (SAS Institute, Cary, NC, USA).

## 3. Results
### (a) Symbionts are smaller, but maintain photosynthetic oxygen production at high densities by more efficient light utilization
Symbionts suffering from potential self-shading in high-density colonies adjusted their photosynthesis towards improved light harvesting and utilization efficiency. This was observed as consistent increase in chlorophyll content, a shift in the relative ratio between primary (chl $a$) and accessory chlorophyll (chl $c_2$) towards a larger antenna, and an increase in maximum quantum yield ($F_v/F_m$) and electron transport rate (rETR$_{max}$) through photosystem II (figure 1$a$–$f$). As a result, photosynthetic oxygen production per symbiont cell was not significantly different across the tissue density range of approximately 1–4 million symbiont cells per milligram of host protein. The observed increased gross oxygen production in colonies with higher symbiont densities is thus a direct linear cumulative effect of the number of symbionts in the tissue (figure 1$g$,$h$).

Regular feeding led the coral colonies to appear visually darker as result of increased areal symbiont density and symbiont chlorophyll content as well as a higher host protein per surface area content (all + 25–28%; figure 1$e$; electronic supplementary material, figure S3A, B and table S1). Note that while most replicates saw an increase in symbiont density and areal host protein content, the overall effect was not significant ($p = 0.0539$ for both). Importantly, the number of symbionts and the amount of host tissue per unit surface area both increased isometrically, which maintained the relative symbiont density per amount of host tissue at a similar level between fed and unfed corals (electronic supplementary material, figure S3A–C and table S1). The symbiont response to feeding-induced alterations in the light microenvironment was similar to their response to increased density: symbionts in fed colonies had higher pigment content and light utilization efficiency ($F_v/F_m + 2.4\%$, rETR$_{max} + 24\%$ and $I_k + 29\%$; figure 1$a$–$d$). By contrast, their individual $O_2$-productivity (−33%) was consistently lower across the whole density range, thus lowering $P_{gross}$ (−23%) and $P_{net}$ (−29%) per amount of host tissue (figure 1$g$–$j$; electronic supplementary material, table S1). However, the tendency for more coral protein per surface area in fed corals kept the colony's areal gross oxygen production unaltered between both feeding treatments (electronic supplementary material, table S1). We found no evidence that feeding-induced increases in density and host tissue amplified the self-shading effects seen at the high end of the density spectrum. However, the density gradient strongly correlated with the symbiont's biomass. Independent of the holobiont's feeding state, symbiont soluble carbohydrate and protein content per cell declined by almost 90% towards the highest density in a hyperbolic fashion (figure 2; electronic supplementary material, table S2A). No shift in the relative ratio between carbohydrate and protein content in the symbionts was observed (data not shown).

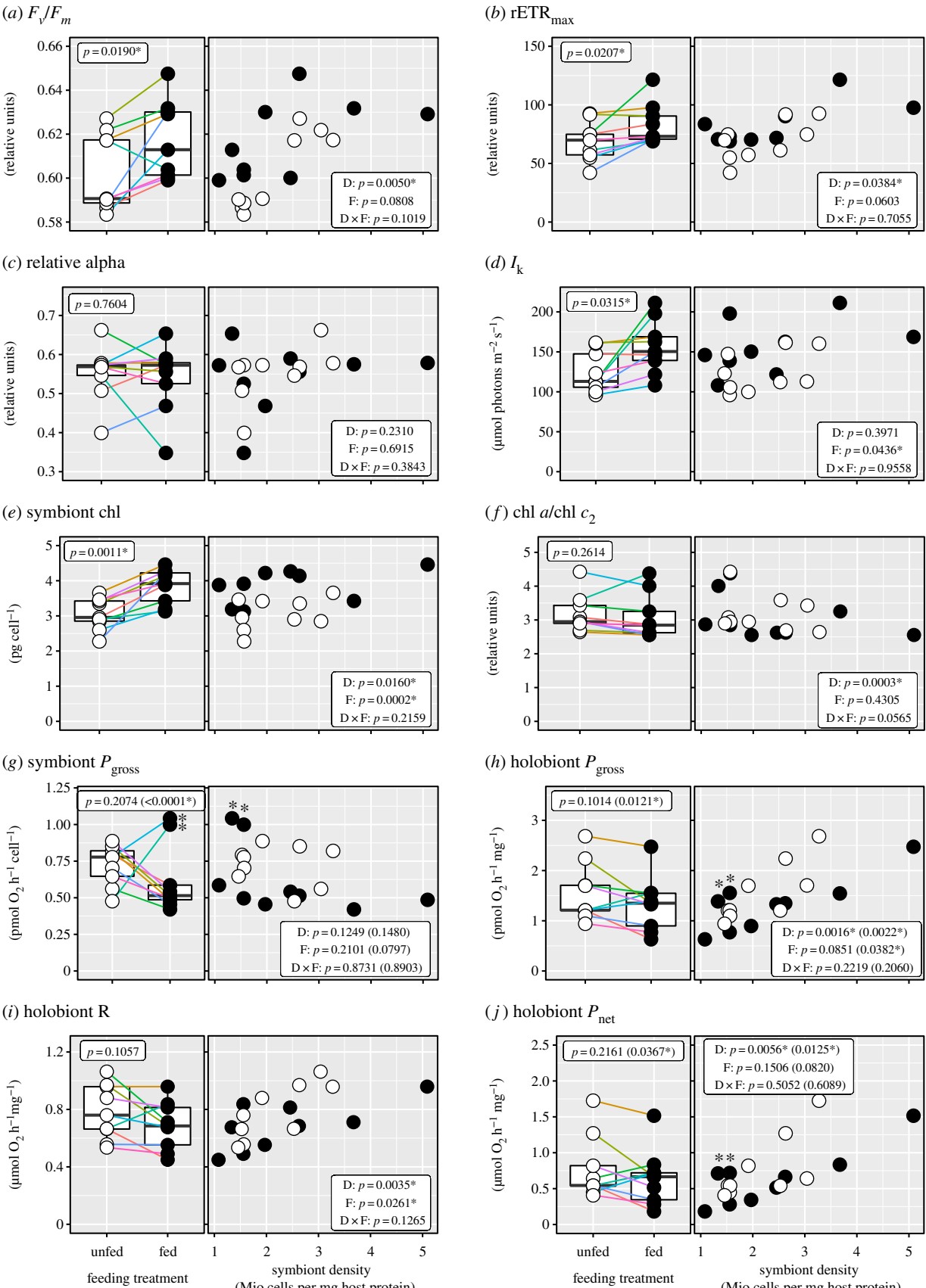

**Figure 1.** Coral symbiont density affects dinoflagellate photophysiology. Boxplots show effects of regular feeding (unfed [white], fed [black]) on physiological variables in paired fragments (colours) of the *Stylophora pistillata* population ($n = 9$). Scatter plots show the same physiological variables in relation to the absolute symbiont density in the host tissue of each colony. (*a*–*f*) Photophysiological properties of the symbiont: (*a*) Maximum quantum yield of photosystem II, (*b*) relative maximum electron transport rate of photosystem II, (*c*) relative initial slope of the light curve, (*d*) minimum photosynthetic saturation irradiance, (*e*) symbiont chlorophyll (chl) content, (*f*) ratio between primary and accessory pigment. (*g*–*j*) Oxygen productivity of the coral colony with regard to (*g*) individual symbiont and (*h*) holobiont gross oxygen production, (*i*) holobiont respiration and (*j*) holobiont net oxygen release. Shown statistical details refer to the average feeding effect (boxplots; electronic supplementary material, table S1) and the effects of density (D) and feeding (F) (electronic supplementary material, table S3); $n = 9$, except for *g*, *h*, *j* where the statistical results are shown including and excluding two outlier colony pairs (highlighted with *). (Online version in colour.)

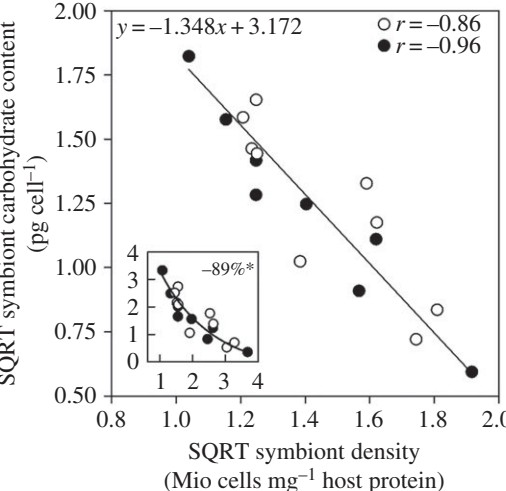
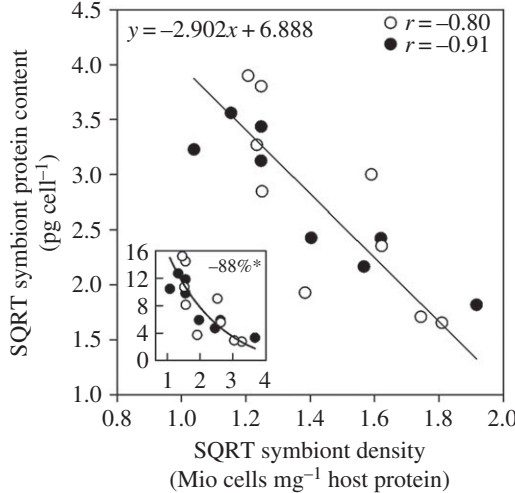

**Figure 2.** Dinoflagellate cell biomass declines as a function of their density in the coral tissue. Coral colonies show a significant negative correlation between the average tissue symbiont density and the soluble protein and carbohydrate of their symbiont cells, independent of their feeding state (unfed, white; fed, black). Inserts display the original hyperbolic relationship between density and biomass. Indicated reductions (in %) over the observed density range are derived from the shown orthogonal regressions (electronic supplementary material, table S2A).

### (b) Symbiont density does not alter host protein or carbohydrate content, but controls host antioxidant enzyme activity

The number of coral symbionts did not affect the amount of protein or carbohydrate in the host tissue in either feeding treatment (electronic supplementary material, figure S3D-E). We observed significant increases in the activity of host antioxidant enzymes related to the scavenging of superoxide and hydrogen peroxide in both feeding regimes (electronic supplementary material, figure S3F–G). On the symbiont side, only superoxide dismutase (SOD) activity showed a positive response to density, but only in unfed corals that had a higher overall oxygen production compared with the fed treatment. Symbiont catalase peroxidase activity (KatG) was not responsive to density changes (electronic supplementary material, figure S3H–I, tables S1 and S3). Holobiont respiration (i.e. symbiont + host) showed a positive response to the tissue symbiont density (figure 1$i$; electronic supplementary material, table S3), but it is not possible to interpret this in a meaningful way as this value encompasses both partners. Thus, we cannot disentangle potential changes in host respiration from the increased holobiont respiration that is due to the presence of a higher number of symbionts per unit host tissue.

### (c) Density alters symbiont and host nutrient assimilation

In addition to the colony-scale view obtained with standard physiological measurements, NanoSIMS allowed us to observe density effects on the scale of individual symbiont cells embedded in the host tissue. These data showed a statistical significant decline of *in hospite* dinoflagellate assimilation for nitrate (−23%), but not dissolved inorganic carbon (−12%), over the density range of 1–9 symbiont cells per 500 µm$^{-2}$ of gastrodermal cross-sectional area (figure 3$a$; electronic supplementary material, tables S2B and S4A), with no effect of feeding. Similar effects were observed for the assimilation of recycled host metabolic 'waste' derived from the digestion of dual isotopically labelled zooplankton in regularly fed corals. Density significantly affected $^{15}$N-ammonia and/or dissolved organic nitrogen assimilation (−31%), but not assimilation of

catabolic $^{13}$CO$_2$ (−10%) (figure 3$b$; electronic supplementary material, tables S2C and S4B). In summary, heterotrophic feeding did not affect the observed correlations between local density and symbiont or host carbon and nitrogen assimilation (figure 3).

Despite the reduction in individual symbiont anabolism, host gastrodermal assimilation of the cumulative translocated metabolites increased proportionally with symbiont density (figure 4; electronic supplementary material, tables S2D and S4C). We found some indication that the host tissue in high-density patches assimilated even more autotrophic carbon from the same number of symbiont cells when the host was acclimated to regular heterotrophic feeding (see detailed replicate data in electronic supplementary material, figure S4), but this effect was not statistically significant ($p = 0.0646$).

## 4. Discussion

### (a) Symbionts adjust their photophysiology at higher densities but ultimately compete for nitrogen

Symbiont density had a profound impact on physiological and metabolic variables of the coral holobiont in this study, regardless of feeding status. Adjustments in symbiont pigment content and light utilization sustained the photosynthetic oxygen production per symbiont cell over the observed density range, compensating for the darker microenvironment at the high end of the density spectrum. However, anabolic nutrient assimilation and cell biomass decreased with density, matching previous observations of smaller cells with a lower division rate [10]. Our *in hospite* data provide direct quantitative evidence for increasing exploitation competition between symbiont cells with increasing local tissue symbiont cell density. This increasing competition for nitrogen probably creates a negative feedback for population growth. Thus, the steady-state population at its maximum carrying capacity is indeed characterized by a low nitrogen assimilation and a low generation turnover as conceptualized previously [9]. In agreement with recent findings on the role of phosphorus as another limiting element for symbiont growth and ultrastructure [22,49,50], a NanoSIMS image of the phosphorus distribution in the coral

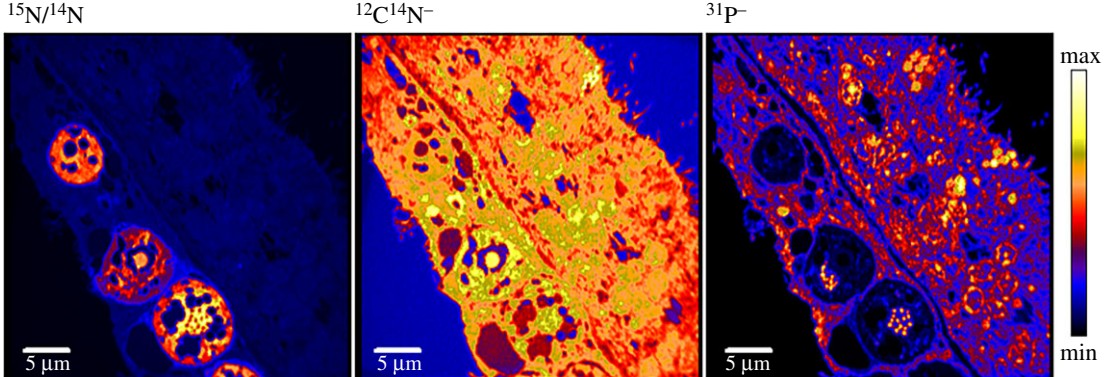

**Figure 3.** Local tissue symbiont density affects symbiont assimilation of carbon and nitrogen. Relative structural carbon and nitrogen turnover in individual dino-flagellates from the assimilation of (*a*) ¹³C-bicarbonate and ¹⁵N-nitrate after 6 h in the light and (*b*) recycled nutrients 6 h after host feeding of dual labelled *Artemia* sp. prey as a function of local tissue symbiont density. Density effects are independent of the host's feeding state in (*a*). The shown regression fits (black lines; electronic supplementary material, table S4) incorporate feeding regime (in (*a*) white dots: unfed, black dots: fed) and replicate-specific behaviour (see electronic supplementary material, figure S4). The average relative change in individual assimilation from 1 to 9 cells per 500 μm² is indicated (electronic supplementary material, tables S2 and S4). Significant effects are highlighted by asterisks.

**Figure 4.** Autotrophic carbon and nitrogen assimilation of the host gastrodermal tissue in relation to its local dinoflagellate density. Relative structural carbon and nitrogen turnover of the host gastrodermis from assimilation of ¹³C-bicarbonate and ¹⁵N-nitrate after 6 h in the light as a function of its local tissue symbiont density. Density effects are independent of the host's feeding regime. The shown regression fit (black line; electronic supplementary material, table S4) incorporates feeding regime (white dots: unfed, black dots: fed) and replicate-specific behaviour (see electronic supplementary material, figure S4). The average relative change in assimilation from 1 to 9 cells per 500 μm² is indicated (electronic supplementary material, tables S2 and S4). Asterisks indicate statistical significance.

$^{15}N/^{14}N$    $^{12}C^{14}N^-$    $^{31}P^-$

**Figure 5.** Low phosphorus abundance in coral symbionts relative to the surrounding host tissue. Dinoflagellate cells (highlighted in $^{15}N/^{14}N$ image) are indistinguishable via NanoSIMS with regard to their nitrogen content ($^{12}C^{14}N^-$), but show very low phosphorus abundance in their tissue compared to their surrounding host tissue. Phosphorus-rich spots inside of symbiont cells are chromosomes. Note that the other nitrogen and phosphorus-deficient spots in the host gastrodermis are lipid bodies. (Online version in colour.)

tissue reveals furthermore that *in hospite* symbionts appear to be severely phosphorus deficient relative to their surrounding host tissue (figure 5). Indeed, the highest cellular phosphorus abundance was observed in the permanently condensed chromosomes, which, considering that Symbiodiniaceae maintain large genomes (approx. 0.7–1.5 Gbp), probably represent a major sink for this element.

Given the symbiont's intracellular location (sequestered in a host vacuole), supply of external inorganic nutrients from the seawater as well as access to internal host metabolic by-products is facilitated solely by the host cell across its symbiosome membrane. The emergence of nutrient competition and the similar degree of reduction in assimilation of external and internal nitrogen sources at high local symbiont densities, point to the supply capacity of the host as another regulatory element in this process. Under the assumption that the supply capacity of external seawater nutrients from host to symbiont has an upper limit, we expected an amplification of symbiont nutrient limitation in corals with an enlarged symbiont community in the feeding treatment. However, we suspect that the isometric increase between host and symbiont biomass in our experiment probably sustained the average supply capacity per symbiont. Further testing of the role and limits of host nutrient supply in a manipulated system where the true load of symbionts per unit host tissue is elevated would be valuable.

In the case of carbon assimilation, there is ambivalent data from different corals on whether *in hospite* symbiont photosynthesis is always carbon-limited [51–56]. Also note that these studies measured oxygen production in photosystem II rather than actual carbon fixation in the Calvin cycle. Our data provide direct evidence that symbiont assimilation of host catabolic carbon, released from digested zooplankton prey, is not limiting at any symbiont density. Although similar results were obtained for the assimilation of seawater DIC, a conclusive verdict on density-dependent autotrophic carbon limitation cannot be drawn, because for practical reasons seawater DIC concentration was doubled in the used 30 l incubation volume here (rather than completely replacing it with DI$^{13}$C).

### (b) Intraspecific symbiont density effects can be as important as symbiont identity

While not affecting evolutionary fitness of symbionts in most coral colonies that harbour a single symbiont genet (i.e. a single clonal line of one species) [57], lower assimilatory performance as well as reduced cell size and generation turnover at high local densities will act as a selective pressure in genetically diverse holobionts. The 20–30% reduction in individual nitrogen assimilation at high density is substantial and of the same magnitude as hosting a completely different symbiont type [44,58]. Density-related competitive effects therefore can contribute to shaping the genetic structure of the symbiont community in colonies with multiple clonal lines of a single symbiont species or with multiple different species. Our data support symbiont density as important factor that determines the functional response of a holobiont to changing environmental conditions, such as increasing temperature or micronutrient availability [59]. Variability in individual assimilation due to the number of cells present should also be considered from a methodological point of view. NanoSIMS studies that intend to compare enrichments across treatments or between symbiont species should be attentive of the potential impact of density on measured enrichment in both partner compartments, especially when the dataset is based on a small number of NanoSIMS images.

### (c) Host anabolic C and N demand are met independent of symbiont behaviour

Symbiont density and intracellular competition had a very limited impact on the measured host variables. Indeed, host carbohydrate and protein content showed no correlation with symbiont density on the colony level and host assimilation of phototrophic carbon and nitrogen increased proportionally with the number of symbiont cells on the local tissue level. It shows that host anabolic turnover of C and N is directly responsive to the number of symbionts even at high local densities. The altered nutrient assimilation of individual symbionts, despite meeting host demand for C and N, revives a previous suggestion that population size and individual nutrient state might affect the quality (i.e. C : N ratio), rather than quantity of photosynthates [60]; a question suited for time-of-flight secondary ion mass spectrometry (ToF-SIMS) on the host tissue in high-density patches. It should be noted that despite the fluctuations in local nutrient assimilation due to density, the potential for long-range (at the scale of centimetres) as well as directed transport to regenerating tissue are macroscale phenomena that demonstrate that the host is not exclusively influenced by localized input [61–63].

The previous suggestion about an optimal symbiont density to meet the metabolic demands of their coral host [64] was not confirmed here. The concept of a 'metabolic optimum' [64] still has insufficient data support, mainly due to the limited number of measured colonies and due to the lack of a clear differentiation between oxygen production and carbon assimilation when relating photosynthesis to density. The crucial and yet unresolved issue is whether coral gross photosynthesis responds linearly (this study) [12,65] or asymptotically [66,67] to symbiont density. Only the latter case creates a density threshold above which cumulative oxygen production (and presumably symbiont carbon delivery) is not keeping up with the increase in holobiont respiration. In *Stylophora pistillata*, we did not find such a threshold for oxygen production on the colony level or host carbon assimilation on the local tissue level. The well-known stimulating effect that cumulative oxygen production has on host antioxidant activity [68,69] was confirmed here. In the context of the broader discussion on density versus productivity, the effect of rising oxygen tensions with density should also be investigated with regard to their negative feedback on carbon fixation in the Calvin cycle. Photorespiratory effects in Symbiodiniaceae are with the exception of a few statements [42,70–72] largely understudied, but strongly relevant for linking oxygen production to actual carbon fixation.

## 5. Conclusion

This study provides direct evidence for the importance of nitrogen limitation and resulting nutrient competition of intracellular symbionts as mechanism for population control in symbiotic corals. While adjusting photosynthesis and maintaining individual oxygen production and (potentially) carbon

assimilation over a large density range, a reduction in symbiont nitrogen assimilation at high tissue densities decreases cell biomass/size and determines the previously observed decline in symbiont division rates when approaching the maximum carrying capacity [9,10]. Despite raising host antioxidant defences, harbouring a larger symbiont population does not change the metabolic benefit for the coral host; potential shifts in the quality of released photosynthates notwithstanding. Stable symbiont photosynthesis and density-proportional assimilation of phototrophic carbon by the host suggest a functional match between demand and supply of inorganic carbon across the range of observed tissue symbiont densities in *Stylophora pistillata* under ambient conditions. In a system where symbiont population growth follows the law of the minimum and host benefits are linked to the density-dependent harvesting of the resulting excess photosynthetic carbon, symbiont competitiveness in assimilating limiting nutrients becomes a key element for defining the balance within the symbiotic relationship.

Data accessibility. All physiological and NanoSIMS data are provided in electronic supplementary material, files.

Authors' contributions. T.K., N.H., J.B. and M.F. designed the experiment. T.K., N.H., J.B. and M.-E.G. carried out physiological and molecular measurements. T.K. conceptualized and conducted the data analysis and wrote the original draft. All authors contributed to editing the manuscript.

Competing interests. The authors declare no competing interests.

Funding. This work was supported by a Swiss National Science Foundation grant no. CR32I2 159282 to A.M. The Red Sea Simulator was funded by an Israel Science Foundation grant to M.F.

Acknowledgements. The manuscript has benefited from feedback of various members of the Laboratory for Biological Geochemistry, from Prof. Isabelle Domart-Coulon, Dr Stephane Roberty and the constructive reviews of two referees.

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
