## [Reviewer comments · Proceedings of the Royal Society B: Biological Sciences]

Review History

RSPB-2019-2344.R0 (Original submission)

Review form: Reviewer 1 (Luke Morris)

Recommendation

Accept with minor revision (please list in comments)

Scientific importance: Is the manuscript an original and important contribution to its field?

Good

General interest: Is the paper of sufficient general interest?

Acceptable

Quality of the paper: Is the overall quality of the paper suitable?

Good

Is the length of the paper justified?

Yes

Should the paper be seen by a specialist statistical reviewer?

Yes

Do you have any concerns about statistical analyses in this paper? If so, please specify them explicitly in your report.

No

It is a condition of publication that authors make their supporting data, code and materials available - either as supplementary material or hosted in an external repository. Please rate, if applicable, the supporting data on the following criteria.

Is it accessible?

Yes

Is it clear?

Yes

Is it adequate?

Yes

Do you have any ethical concerns with this paper?

No

Comments to the Author

I have attached comprehensive comments and recommendations which I hope are useful and constructive in improving the manuscript.

Review form: Reviewer 2 (Daniel Nielsen)

Recommendation

Accept with minor revision (please list in comments)

Scientific importance: Is the manuscript an original and important contribution to its field?

Excellent

General interest: Is the paper of sufficient general interest?

Good

Quality of the paper: Is the overall quality of the paper suitable?

Good

Is the length of the paper justified?

Yes

Should the paper be seen by a specialist statistical reviewer?

No

Do you have any concerns about statistical analyses in this paper? If so, please specify them explicitly in your report.

No

It is a condition of publication that authors make their supporting data, code and materials available - either as supplementary material or hosted in an external repository. Please rate, if applicable, the supporting data on the following criteria.

Is it accessible?

Yes

Is it clear?

Yes

Is it adequate?

Yes

Do you have any ethical concerns with this paper?

No

Comments to the Author

In the presented study, Krueger et al. employed NanoSIMS image data to investigate symbiont density dependent carbon and nitrogen turnover in individual coral symbiont cells as well as in adjacent host tissue. They show that while increasing symbiont density didn't affect the gross photosynthetic output per symbiont due to concurrent adjustments to the photosynthetic machinery, it did result in reduced N turnover on a per symbiont cell basis. The authors conclude that this reduction in N turnover indicates a nitrogen limitation resulting from competition for resources between the individual symbionts, which form a negative feedback and control mechanism on symbiont population growth in the host tissue.

While the experimental work presented in this study has been published previously, the authors are open about this and clearly describe how further analyses of previous image data opened for interpretations of their data in a new context, suitable for a new publication. I find the argument for this sound and complement the authors for looking deeper into their hard-gained image data.

Overall, I find the paper well written with clear arguments that is supported by the presented data. The statistical analyses seem well founded and appropriately conservative. As such, I only have some minor comments below that I hope may further improve some aspects of the paper.

Minor comments

L. 49-56. Please supply references/evidence for the presented argument that photosynthate release by the dinoflagellates in corals is indeed a result of nitrogen limitation and/or nutrient imbalance.

L. 93-96. Since this data is obtained from previous experiments with separate aims, the statement that regulation of host feeding was done to manipulate symbiont densities is not correct in principle. I suggest the wording is changed to reflect that data was obtained that contained this information specifically, rather than produced to investigate these new questions.

L. 105-106. Same as above. The argument that feeding was done to manipulate cell densities is not correct. Instead, it could be explained that the feeding treatment in the previous study turned out to affect the symbiont density, which enabled this study.

L. 212. "host feeding tended to increase the amount of host protein per surface area". From figure S3B (and S3E as referred to) the increase in host areal protein is not significant. It is not clear to me how above statement is to be understood. If this data is not significant, make sure to highlight it in the statement, such as pointing out that most replicates saw an increase in host protein, but the overall effect was not significant.

L. 252. It would be interesting to include a bit of discussion on why it could be that the symbiont

gross photosynthesis was negatively affected by host feeding, since this seems opposite to expectations given the increase in nitrogen input to the holobiont. If this was discussed in a previous publication, then simply refer to this.

L. 262-265. "While not equally...". This sentence feels ambiguous since it seems to indicate NanoSIMS is not appropriate for phosphorous measurements and yet proceeds to conclude on such measurements. Consider clarifying this section for the non-NanoSIMS user - or leave out the technicality all together.

L. 267. Remove comma after "both".

L. 266-268. I find this argument a bit hard to follow. Consider revising to clarify, and in particular what is meant by the host as a "common delimiter in the supply capacity towards its symbionts".

L. 284-297. This paragraph feels like somewhat of a departure from the main tenant of the paper. While it is an interesting point, consider shortening to strengthen the focus through the discussion.

L. 331. "...and a slow down in cell division rates..". As far as I can see, mitotic index was not measured in this study (data is not presented). If this statement relate to data in a previous publication please specify. Otherwise I suggest to tone down the statement so that it is clear this is argument is not based on data from this study.

L. 332. "Although high density triggers..". Add "symbiont" after "high".

Figure 5 is not mentioned in results and feels mostly unrelated to the study. Based on the current focus of the paper and in keeping with the length restriction of the journal, I suggest this figure is moved to the supplementary data.

Decision letter (RSPB-2019-2344.R0)

13-Dec-2019

Dear Dr Krueger:

I am writing to inform you that your manuscript RSPB-2019-2344 entitled "Intracellular competition for nitrogen controls dinoflagellate population density in corals" has, in its current form, been rejected for publication in Proceedings B.

This action has been taken on the advice of referees, who have recommended that substantial revisions are necessary. With this in mind we would be happy to consider a resubmission, provided the comments of the referees are fully addressed. However please note that this is not a provisional acceptance.

Please find below the comments made by the referees, not including confidential reports to the Editor, which I hope you will find useful. While you will see some positive comments made, substantial improvements and constructive responses are required to various elements of your

work. These include parts of the experimental design, as well as assumptions made that relate to your inferences. I would like to point out that, as indicated below, your manuscript will go out for full peer review once more. Please note, however, that it is unlikely to be considered further, unless there is a substantial improvement and recognition by the referees, board member and myself, that your manuscript is likely to command the quality and expectations of impact required. If you do choose to resubmit your manuscript, please upload the following:

Sincerely,
 Professor Gary Carvalho
 mailto: proceedingsb@royalsociety.org

Associate Editor
 Board Member: 1
 Comments to Author:
 Dear Dr Krueger

I have received two reviews of your manuscript. Both reviewers make a number of comments that need to be addressed before the ms can be considered further. These include a broadening of the discussion to include corals from the high treatment described. I look forward to a revised ms with a cover letter that describes a point by point response to all comments.

Best
 Line K Bay

Reviewer(s)' Comments to Author:

Referee: 1

Comments to the Author(s)

I have attached comprehensive comments and recommendations which I hope are useful and constructive in improving the manuscript.

Referee: 2

Comments to the Author(s)

In the presented study, Krueger et al. employed NanoSIMS image data to investigate symbiont density dependent carbon and nitrogen turnover in individual coral symbiont cells as well as in adjacent host tissue. They show that while increasing symbiont density didn't affect the gross photosynthetic output per symbiont due to concurrent adjustments to the photosynthetic machinery, it did result in reduced N turnover on a per symbiont cell basis. The authors conclude that this reduction in N turnover indicates a nitrogen limitation resulting from competition for

resources between the individual symbionts, which form a negative feedback and control mechanism on symbiont population growth in the host tissue.

While the experimental work presented in this study has been published previously, the authors are open about this and clearly describe how further analyses of previous image data opened for interpretations of their data in a new context, suitable for a new publication. I find the argument for this sound and complement the authors for looking deeper into their hard-gained image data.

Overall, I find the paper well written with clear arguments that is supported by the presented data. The statistical analyses seem well founded and appropriately conservative. As such, I only have some minor comments below that I hope may further improve some aspects of the paper.

Minor comments

L. 49-56. Please supply references/evidence for the presented argument that photosynthate release by the dinoflagellates in corals is indeed a result of nitrogen limitation and/or nutrient imbalance.

L. 93-96. Since this data is obtained from previous experiments with separate aims, the statement that regulation of host feeding was done to manipulate symbiont densities is not correct in principle. I suggest the wording is changed to reflect that data was obtained that contained this information specifically, rather than produced to investigate these new questions.

L. 105-106. Same as above. The argument that feeding was done to manipulate cell densities is not correct. Instead, it could be explained that the feeding treatment in the previous study turned out to affect the symbiont density, which enabled this study.

L. 212. "host feeding tended to increase the amount of host protein per surface area". From figure S3B (and S3E as referred to) the increase in host areal protein is not significant. It is not clear to me how above statement is to be understood. If this data is not significant, make sure to highlight it in the statement, such as pointing out that most replicates saw an increase in host protein, but the overall effect was not significant.

L. 252. It would be interesting to include a bit of discussion on why it could be that the symbiont gross photosynthesis was negatively affected by host feeding, since this seems opposite to expectations given the increase in nitrogen input to the holobiont. If this was discussed in a previous publication, then simply refer to this.

L. 262-265. "While not equally...". This sentence feels ambiguous since it seems to indicate NanoSIMS is not appropriate for phosphorous measurements and yet proceeds to conclude on such measurements. Consider clarifying this section for the non-NanoSIMS user - or leave out the technicality all together.

L. 267. Remove comma after "both".

L. 266-268. I find this argument a bit hard to follow. Consider revising to clarify, and in particular what is meant by the host as a "common delimiter in the supply capacity towards its symbionts".

L. 284-297. This paragraph feels like somewhat of a departure from the main tenant of the paper. While it is an interesting point, consider shortening to strengthen the focus through the discussion.

L. 331. "...and a slow down in cell division rates.". As far as I can see, mitotic index was not measured in this study (data is not presented). If this statement relate to data in a previous publication please specify. Otherwise I suggest to tone down the statement so that it is clear this is argument is not based on data from this study.

L. 332. "Although high density triggers..". Add "symbiont" after "high".

Figure 5 is not mentioned in results and feels mostly unrelated to the study. Based on the current focus of the paper and in keeping with the length restriction of the journal, I suggest this figure is moved to the supplementary data.

Author's Response to Decision Letter for (RSPB-2019-2344.R0)

See Appendix A.

RSPB-2020-0049.R0

Review form: Reviewer 1 (Luke Morris)

Recommendation

Accept as is

Scientific importance: Is the manuscript an original and important contribution to its field?

Excellent

General interest: Is the paper of sufficient general interest?

Good

Quality of the paper: Is the overall quality of the paper suitable?

Excellent

Is the length of the paper justified?

Yes

Should the paper be seen by a specialist statistical reviewer?

No

Do you have any concerns about statistical analyses in this paper? If so, please specify them explicitly in your report.

No

It is a condition of publication that authors make their supporting data, code and materials available - either as supplementary material or hosted in an external repository. Please rate, if applicable, the supporting data on the following criteria.

Is it accessible?

Yes

Is it clear?

Yes

Is it adequate?

Yes

Do you have any ethical concerns with this paper?

No

Comments to the Author

The authors have addressed the constructive criticisms and significantly improved the quality of the manuscript. I would recommend that the revised manuscript be accepted for publication.

Some specific comments regarding the arguments for not including temperature in the manuscript:

1. Even though physiology was not impacted by temperature in these corals, other important manuscripts have been published where physiology was not been impacted temperature but carbon and nitrogen metabolism were. These manuscripts, including those be the authors (and others such as Baker et al 2018, Gibbin et al 2018) made important and novel contributions. I don't think it would be misleading to publish this data as long as it is made clear that these are corals with higher thermal tolerance and it is important to know how the nutrient metabolism of Red Sea corals works in contrast to those sensitive to climate change.

2. I can agree that it would complicate to model to include temperature in addition to feeding. It would still be valuable to publish an analysis where temperature is included and feeding excluded (perhaps in a different manuscript), even if they are null results. However, I realise you can make the same arguments to published the analysis involving feeding, rather than temperature, so this manuscript should still be published regardless.

Review form: Reviewer 2 (Daniel Nielsen)

Recommendation

Accept as is

Scientific importance: Is the manuscript an original and important contribution to its field?

Excellent

General interest: Is the paper of sufficient general interest?

Excellent

Quality of the paper: Is the overall quality of the paper suitable?

Excellent

Is the length of the paper justified?

Yes

Should the paper be seen by a specialist statistical reviewer?

No

Do you have any concerns about statistical analyses in this paper? If so, please specify them explicitly in your report.

No

It is a condition of publication that authors make their supporting data, code and materials available - either as supplementary material or hosted in an external repository. Please rate, if applicable, the supporting data on the following criteria.

Is it accessible?

Yes

Is it clear?

Yes

Is it adequate?

Yes

Do you have any ethical concerns with this paper?

No

Comments to the Author

I commend the authors on their thorough revisions and in particular of the improvements to the discussion.

I have no further comments and recommend the paper for acceptance.

Decision letter (RSPB-2020-0049.R0)

28-Jan-2020

Dear Dr Krueger

I am pleased to inform you that your manuscript RSPB-2020-0049 entitled "Intracellular competition for nitrogen controls dinoflagellate population density in corals" has been accepted for publication in Proceedings B.

The referee(s) have recommended publication, but also suggest some minor revisions to your manuscript. Therefore, I invite you to respond to the referee(s)' comments and revise your manuscript. Because the schedule for publication is very tight, it is a condition of publication that you submit the revised version of your manuscript within 7 days. If you do not think you will be able to meet this date please let us know.

Sincerely,
 Professor Gary Carvalho
 mailto: proceedingsb@royalsociety.org

Associate Editor
 Comments to Author:
 Dear Dr Krueger

I have received assessments for the two original reviewers of your ms. They are both satisfied with your responses and the changes you made to the ms. I have read through the final, track-change version and cover letter and have come to the same conclusion.

I found a few minor areas that should be addressed in a final version. In all - a very interesting paper of general interest.

Best
 Line K Bay

L245: I suggest to include a "Here we did" to make it clear to the reader you are talking about your results specifically.

L266 perhaps add "...of this process."

L267: a reference would be appropriate

L272 the text after the ";" is a bit disjointed. I would revise

L296: Was this expected based on literature ?

I agree w Reviewer 1 that it would be worth mentioning that this is an extremely heat tolerant coral - perhaps in the first few lines of M and M

L337: Lastly - given you acknowledge input from lab - perhaps it would be appropriate to also ack. the reviewers?

Reviewer(s)' Comments to Author:

Referee: 1

Comments to the Author(s).

The authors have addressed the constructive criticisms and significantly improved the quality of the manuscript. I would recommend that the revised manuscript be accepted for publication.

Some specific comments regarding the arguments for not including temperature in the manuscript:

1. Even though physiology was not impacted by temperature in these corals, other important manuscripts have been published where physiology was not been impacted temperature but carbon and nitrogen metabolism were. These manuscripts, including those by the authors (and others such as Baker et al 2018, Gibbin et al 2018) made important and novel contributions. I don't think it would be misleading to publish this data as long as it is made clear that these are corals with higher thermal tolerance and it is important to know how the nutrient metabolism of Red Sea corals works in contrast to those sensitive to climate change.

2. I can agree that it would complicate to model to include temperature in addition to feeding. It would still be valuable to publish an analysis where temperature is included and feeding excluded (perhaps in a different manuscript), even if they are null results. However, I realise you

can make the same arguments to published the analysis involving feeding, rather than temperature, so this manuscript should still be published regardless.

Referee: 2

Comments to the Author(s).

I commend the authors on their thorough revisions and in particular of the improvements to the discussion.

I have no further comments and recommend the paper for acceptance.

Decision letter (RSPB-2020-0049.R1)

06-Feb-2020

Dear Dr Krueger

I am pleased to inform you that your manuscript entitled "Intracellular competition for nitrogen controls dinoflagellate population density in corals" has been accepted for publication in Proceedings B.

Open Access

Paper charges

Sincerely,
Proceedings B
[mailto: proceedingsb@royalsociety.org](mailto:proceedingsb@royalsociety.org)

Appendix A

Response to referees

We thank both reviewers for their constructive criticism and suggestions on how to improve the manuscript.

Reviewer 1: Luke Morris

The manuscript addresses the topical question of how algal symbiont density in tropical corals is regulated, and how it impacts the health, productivity and nutrient metabolism. Previous studies (hypothetical and empirical) have provided mixed results, some finding that high density symbiont populations become a burden upon their coral hosts and increase the susceptibility of corals to thermal bleaching, whereas others find no or opposite effects. This study therefore provides a valuable and timely empirical integration of physiological and metabolic data in an attempt to resolve this.

In general, the manuscript is of high quality and the introduction, methods and results are well written and presented. I have made some recommendations for minor revisions to these sections. I also highly recommend that if at all possible, the same results are presented for the corals acclimated to high temperature in the same experiment. This data could contribute an extremely valuable insight to the field of coral nutrient metabolism and stress tolerance and would improve the impact of the manuscript.

In terms of the discussion and conclusion, the interpretation is well thought out, but I would recommend edits to bring the writing up to the same quality as the rest of the manuscript.

Comments on introduction

In general, the introduction is very well written and provides an appropriate overview of the study area. I however feel that a couple of generalisations are made about the links between nutrient availability, symbiont density and coral performance which do not necessarily reflect the available evidence in its entirety.

Line 57: This paragraph states the elevated nutrients increase symbiont growth in “many (but not all) studies”. However, it is concluded that “the coral host is incapable of preventing symbiont population growth”. I would take more caution in reaching this conclusion. As you acknowledge, not all studies reach these findings. In particular, the impacts of nutrient enrichment on symbiont growth may be highly dependent on nutrient form and ratio, as highlighted in the meta-analysis of Shantz and Burkepile (2014: <https://doi.org/10.1890/13-1407.1>) and my own recent review which encompasses this subject (Morris et al 2019: <https://doi.org/10.1016/j.tim.2019.03.004>). Regardless, I don't believe the finding that elevated nutrients increase symbiont growth contradicts mechanisms of host control of nutrient supply (e.g. Cui et al 2019: <https://dx.doi.org/10.1371/journal.pgen.1008189>), rather that these control mechanisms can become overwhelmed or adjusted to suit environmental conditions.

Response: We have rephrased the section according to the reviewer's suggestions. With regard to the Cui et al 2019 reference, we only considered coral symbiosis papers in our manuscript. The anemone-Symbiodiniaceae symbiosis is fundamentally different with regard to the nutritional relationship (e.g. a large proteinaceous host providing so much ammonia that there is no detectable symbiont nitrate fixation; no host skeletal carbon sink etc.).

Line 66: This sentence states that higher densities provide more productivity but higher susceptibility to bleaching. However, in reality previous studies on this subject have provided mixed results and productivity and thermal tolerance may be more related to nutrient availability (Morris et al 2019: <https://doi.org/10.1016/j.tim.2019.03.004>). I would suggest that you instead highlight

that this subject remains largely unresolved and controversial and therefore that this study represents a timely and valuable contribution to the field.

Response: Since the purpose of this paragraph is to highlight the methodological challenge of accurately measuring the partner interaction in the in hospite context and since this paper is not about stress phenomena we have removed this sentence altogether to avoid any ambiguity.

Comments on methods

The study clearly presents an additional novel analysis of an experiment which analysed the physiological and metabolic responses of Red Sea corals to heterotrophic feeding and elevated temperature. However, it is notable that the corals exposed to elevated temperature are omitted in this reanalysis. I would highly recommend including results from the high temperature corals in this paper if at all possible, given that the response of corals to ocean warming are such a contemporary topic of general interest. Furthermore, as the links between symbiont density and susceptibility to bleaching are unresolved these results would be very valuable to the field. Although I appreciate that in the Krueger et al. 2017 paper elevated temperature reduced the natural variability in symbiont density and that the corals did not bleach, I feel that the results would still be very informative.

Response: The matter of adding all the high temperature data to this manuscript has been sincerely discussed between the lead authors in the initial stages of the manuscript. However, we came to the conclusion that it would complicate and distort the primary message of this manuscript considerably. There are two main reasons for why the data was not included. While we agree with the reviewer that the nutrient limitation story in a stressed coral would be a topic of interest, the temperature treatment of 11 degree heating weeks (DHW), although reducing symbiont and host carbon and nitrogen assimilation, had no substantial negative impacts on these corals. Instead, it led to an overall improvement of coral photosynthesis (see Krueger et al. 2017). 11 DHW would likely kill most other corals. However in the case of Northern Red Sea corals, traditional temperature bleaching thresholds do not apply (see Fine et al 2013 *Global Change Biology*, Grottoli et al. 2017 *Frontiers*, Bellworthy et al 2017 *Coral Reefs*, Osman et al. 2017 *Global Change Biology*), thus, the presented data would not represent a typical stressed coral phenotype and might be misleading and not transferrable to corals in other parts of the world.

Second, the inclusion of the factor temperature into the statistical analysis of the NanoSIMS data leads to a very complex model that includes a significant three-way interaction for the main fixed factors and a number of additional random interactive factors, making a clear interpretation very difficult. When analysing the ambient and high temperature dataset separately, we found no density effect at the high temperature, but it is impossible to quantitatively relate this to the ambient treatment without a joined statistical model. Given the "special" thermal biology of these corals, it would also be highly speculative and difficult to attribute whether the absence of a density effect is due to actual stress or due to a general metabolic improvement (see Krueger et al. 2017).

The last point we would like to raise is a pragmatic one. Considering that the manuscript in its current form already stretches the length and publishing format permissible for this journal, the inclusion of the high temperature data would effectively double the already extensive amount of physiological and NanoSIMS data that is presented and discussed in this paper.

Line 105: No evidence is presented that the feeding treatment provided "balanced extra nutrition". I suggest that information on the C:N:P ratio of the food is provided if possible along with information on the dissolved nutrients in seawater. It is possible that effects of heterotrophic feeding could extend beyond simply increasing symbiont growth (or not increasing symbiont growth as in this manuscript). The N:P ratio of heterotrophic and autotrophic nutrients could be different and therefore differentially impact coral health and/or heterotrophic feeding may provide little benefit

to corals if inorganic nutrient conditions are stressful. I reviewed this subject (Morris et al 2019: <https://doi.org/10.1016/j.tim.2019.03.004>) but key papers providing potential evidence for this are Rosset et al (2015: <https://doi.org/10.3389/fmars.2015.00103>) and Ezzat et al (2019: <https://doi.org/10.1111/1365-2435.13285>).

Response: We agree with the reviewer on the ambiguity of the term “balanced nutrition” and have removed the whole paragraph. The original intention was to highlight that the heterotrophic supply provided a food supplement that covered a range of elements (e.g. including P) as they would be found in *Artemia* that was exclusively raised by feeding on phytoplankton for nine days. No specific C:N:P analysis on the heterotrophic food was conducted.

Line 166: As the high temperature results were not included in this manuscript, I believe that the inclusion of the factor “temperature” in the text is an error.

Response: Thanks for spotting this. It is indeed an error and a remnant of the early manuscript stage.

Comments on results

In general, I see that p values are reported in some areas of the text but not others. I would recommend being consistent and accurate in the reporting of p values throughout. Otherwise, the results are presented clearly and accurately.

Response: We agree. As the exhaustive statistical output can be viewed in the supplements, we have removed p-values from the manuscript and simply refer to “statistically significant” results were appropriate. For the rare case with borderline non-significant results we have kept the actual p-value.

Line 227: The 12% decrease in carbon assimilation with symbiont density is not significant and it is important to highlight this better. Line 230: Again, the carbon assimilation effect is not significant.

Response: We have rewritten the section accordingly

Line 233: I would clarify that you mean photosynthesis per symbiont cell.

Response: Done

Comments on discussion

Line 245: It appears to me that the same conclusions regarding symbiont density effects can be made regardless of the heterotrophic feeding, and the marginal impacts feeding had on symbiont density. I would instead highlight that there are significant relationships between symbiont density and physiological and metabolic variables and that these hold true regardless of feeding status.

Response: We have rewritten the section accordingly

Line 247: I recommend rephrasing the sentence which starts on this line to improve clarity and readability.

Response: Done

Lines 254-260: I recommend rephrasing these three sentences to improve clarity and readability. They can probably be condensed into one or two sentences whilst still retaining the important information.

Response: Done

Lines 260-265: I would note that high density populations of smaller symbionts are not indicative of severe nutrient stress (for nitrogen or phosphorus), which produces low density populations of enlarged symbionts (Rosset et al 2015: <https://doi.org/10.3389/fmars.2015.00103>; Rosset et al 2017: <https://dx.doi.org/10.1016%2Fj.marpolbul.2017.02.044>). The NanoSIMS visualisation of phosphorus however is novel and informative and highlights previous work noting that corals and especially their symbionts are primarily phosphorus rather than nitrogen limited (e.g. Godinot et al 2011: <https://doi.org/10.1016/j.jembe.2011.08.022>). It would be extremely interesting and informative to see a comparative analysis of phosphorus abundance and distribution between feeding and temperature treatments and in corals with different symbiont density. However, as no comparison is offered it could well be that this phosphorus distribution is typical of corals.

Response: The results from the NanoSIMS visualization of the phosphorus are indeed surprising and amazing. However, considering that phosphorus acquisition in the NanoSIMS is very very time-consuming (low abundance+low ionisation yield), the shown picture is anecdotal and intended to add to the still unfolding story about the importance of phosphorus.

Lines 266-274: I recommend rephrasing and condensing this paragraph to improve clarity and readability. I see that the key point is that any feeding-induced increases in symbiont density were the indirect result of increased host biomass. The conclusion that it would be valuable to repeat the experiment in corals with altered symbiont to host cell ratios is valid.

Response: Done

Line 279: I recommend rephrasing the sentence which starts on this line to improve clarity and readability.

Response: Done

Lines 284-288: I recommend rephrasing these sentences to improve clarity and readability. I see that the key point is that symbiont density is equally as important in determining holobiont fitness as symbiont identity (e.g. Cunning and Baker 2014: <https://doi.org/10.3389/fmicb.2014.00400>).

Response: We have revised this paragraph

Line 311: Also, the idea of an optimal symbiont density is confounded by the availability of resources to support the symbiont population (Morris et al 2019: <https://doi.org/10.1016/j.tim.2019.03.004>). Your study provides potential evidence for high density symbiont potentially being more mutualistic than the low-density populations, due to the higher accumulation of nutrients in the host compartment at the cost of the symbiont. This is the complete opposite of Woodriddle's hypothesis which suggests they are more parasitic. Similar evidence can be found in the literature (e.g. Rosset et al 2015: <https://doi.org/10.3389/fmars.2015.00103>; Rosset et al 2017: <https://dx.doi.org/10.1016%2Fj.marpolbul.2017.02.044>). However, the potential for lower quality (rather than lower quantity) photosynthates is valid and has been found in corals previously under the stressful condition of thermal stress (Hillyer et al 2018: <https://doi.org/10.1007/s11306-017-1306-8>).

Response: The use of labels such as "mutualistic" and "parasitic" are context-dependent in the coral symbiosis and have to be applied to specific aspects of the symbiosis. Thus, we tried to avoid to make a clear judgement in this manuscript. For the nutritional level, the symbionts are beneficial for the host as they provide nutrients to the host. Of course, more symbionts provide cumulatively more nutrients. However, a change in symbiont assimilation rate does not equal parasitism, especially not when host assimilation is unaltered (as was the case of some recent papers that nevertheless proposed parasitism, e.g. Baker et al. 2017 ISME, Radecker et al. 2018 Frontiers). We mention Woodriddle's idea of the metabolic optimum, because in a paper about the effects of symbiont

density it has to be addressed. We hope it becomes clear that in our opinion the question is still open, and is still lacking appropriate data support.

Line 316: I recommend rephrasing the sentence which starts on this line to improve clarity and readability.

Response: Done

Comments on conclusion

The ability of the symbionts to behave in such an “unselfish” way despite being under the stress of competition could be linked to general coral stress tolerance (e.g. they withstood 11 DHW in your paper Krueger et al 2017).

Response: See earlier comment about the stress tolerance of Northern Red Sea corals. While an interesting idea, it seems speculative at this point.

Line 337: I recommend rephrasing the sentence which starts on this line to improve clarity and readability.

Response: We have rephrased the final paragraph.

Referee: 2 Comments to the Author(s)

In the presented study, Krueger et al. employed NanoSIMS image data to investigate symbiont density dependent carbon and nitrogen turnover in individual coral symbiont cells as well as in adjacent host tissue. They show that while increasing symbiont density didn't affect the gross photosynthetic output per symbiont due to concurrent adjustments to the photosynthetic machinery, it did result in reduced N turnover on a per symbiont cell basis. The authors conclude that this reduction in N turnover indicates a nitrogen limitation resulting from competition for resources between the individual symbionts, which form a negative feedback and control mechanism on symbiont population growth in the host tissue.

While the experimental work presented in this study has been published previously, the authors are open about this and clearly describe how further analyses of previous image data opened for interpretations of their data in a new context, suitable for a new publication. I find the argument for this sound and complement the authors for looking deeper into their hard-gained image data.

Overall, I find the paper well written with clear arguments that is supported by the presented data. The statistical analyses seem well founded and appropriately conservative. As such, I only have some minor comments below that I hope may further improve some aspects of the paper.

Minor comments

L. 49-56. Please supply references/evidence for the presented argument that photosynthate release by the dinoflagellates in corals is indeed a result of nitrogen limitation and/or nutrient imbalance.

Response: Given the limitation in manuscript length, the limited number of references, and the fact that this concept has been highlighted in most reviews on the coral symbiosis, we have only referenced the original publication that outlined this concept for the coral symbiosis (Jokiel et al 1994) and a review that discusses this fundamental concept of uncoupling primary production from

population growth in other aquatic systems (Dubinsky & Berman-Frank 2001). Since the law of the minimum (Sprengel/Liebig, 19th century) and the Redfield ratio (Alfred Redfield, early 20th century) are widely known concepts in the field, we did not provide the primary references.

L. 93-96. Since this data is obtained from previous experiments with separate aims, the statement that regulation of host feeding was done to manipulate symbiont densities is not correct in principle. I suggest the wording is changed to reflect that data was obtained that contained this information specifically, rather than produced to investigate these new questions.

Response: We agree and have revised the respective sentences throughout the manuscript.

L. 105-106. Same as above. The argument that feeding was done to manipulate cell densities is not correct. Instead, it could be explained that the feeding treatment in the previous study turned out to affect the symbiont density, which enabled this study.

Response: Done

L. 212. “host feeding tended to increase the amount of host protein per surface area”. From figure S3B (and S3E as referred to) the increase in host areal protein is not significant. It is not clear to me how above statement is to be understood. If this data is not significant, make sure to highlight it in the statement, such as pointing out that most replicates saw an increase in host protein, but the overall effect was not significant.

Response: We have clarified this throughout the paragraph and removed this particular phrase. While statistically not significant at $p=0.0539$, the similar increase in amount of symbiont and host tissue per surface area were ultimately responsible for the fact that no difference in oxygen production per coral surface area between both feeding treatments was found (Table S1), even though gross oxygen per symbiont cell was on average 33% lower. Thus, the (non-significant) 28% increase in areal symbiont density completely compensated for this decline on the colony level in the fed treatment.

L. 252. It would be interesting to include a bit of discussion on why it could be that the symbiont gross photosynthesis was negatively affected by host feeding, since this seems opposite to expectations given the increase in nitrogen input to the holobiont. If this was discussed in a previous publication, then simply refer to this.

Response: We are at a loss about this result. However, it is a consistent observation across pH and temperature treatments across three independent time points in the original experiment (unpublished). While changing the absolute oxygen production per cell, but not per surface area (see response to L212 comments), and not having an influence on the density vs. oxygen production relationship (Fig. 1G), we did not further comment or discuss this in detail as it is of no consequence for the primary messages of the manuscript.

L. 262-265. “While not equally...”. This sentence feels ambiguous since it seems to indicate NanoSIMS is not appropriate for phosphorous measurements and yet proceeds to conclude on such measurements. Consider clarifying this section for the non-NanoSIMS user – or leave out the technicality all together.

Response: We have revised this section accordingly. Please also see our response to Reviewer 1 comment on L260-265.

L. 267. Remove comma after “both”.

Response: Done

L. 266-268. I find this argument a bit hard to follow. Consider revising to clarify, and in particular what is meant by the host as a “common delimiter in the supply capacity towards its symbionts”.

Response: We have revised this section.

L. 284-297. This paragraph feels like somewhat of a departure from the main tenant of the paper. While it is an interesting point, consider shortening to strengthen the focus through the discussion.

Response: We agree and have broken up the paragraph into another subsection.

L. 331. "...and a slow down in cell division rates..". As far as I can see, mitotic index was not measured in this study (data is not presented). If this statement relate to data in a previous publication please specify. Otherwise I suggest to tone down the statement so that it is clear this is argument is not based on data from this study.

Response: We have revised this section.

L. 332. "Although high density triggers..". Add "symbiont" after "high".

Response: Done

Figure 5 is not mentioned in results and feels mostly unrelated to the study. Based on the current focus of the paper and in keeping with the length restriction of the journal, I suggest this figure is moved to the supplementary data.

Response: Considering its relevance to the story about nutrient limitation as well as its uniqueness as an *in hospite* perspective on the abundance of phosphorus inside the symbionts relative to their surrounding host environment, we prefer to keep this small figure in the main text as long as space permits. Please also see our response to Reviewer 1 comment on L260-265.